# Glucose Exerts an Anti-Melanogenic Effect by Indirect Inactivation of Tyrosinase in Melanocytes and a Human Skin Equivalent

**DOI:** 10.3390/ijms21051736

**Published:** 2020-03-03

**Authors:** Sung Hoon Lee, Il-Hong Bae, Eun-Soo Lee, Hyoung-June Kim, Jongsung Lee, Chang Seok Lee

**Affiliations:** 1Amorepacific Corporation R&D Center, Yongin City 17074, Gyunggi-do, Korea; imstrong20@gmail.com (S.H.L.); baelong98@naver.com (I.-H.B.); soopian@amorepacific.com (E.-S.L.); leojune@amorepacific.com (H.-J.K.); 2Department of Integrative Biotechnology, College of Biotechnology and Bioengineering, Sungkyunkwan University, Suwon City 16419, Gyunggi-do, Korea; 3Department of Beauty and Cosmetic Science, Eulji University, Seongnam City 13135, Gyunggi-do, Korea

**Keywords:** melanogenesis, sugar, glucose, tyrosinase, human skin equivalent

## Abstract

Sugars are ubiquitous in organisms and well-known cosmetic ingredients for moisturizing skin with minimal side-effects. Glucose, a simple sugar used as an energy source by living cells, is often used in skin care products. Several reports have demonstrated that sugar and sugar-related compounds have anti-melanogenic effects on melanocytes. However, the underlying molecular mechanism by which glucose inhibits melanin synthesis is unknown, even though glucose is used as a whitening as well as moisturizing ingredient in cosmetics. Herein, we found that glucose significantly reduced the melanin content of α-melanocyte-stimulating hormone (MSH)-stimulated B16 cells and darkly pigmented normal human melanocytes with no signs of cytotoxicity. Furthermore, topical treatment of glucose clearly demonstrated its whitening efficacy through photography, Fontana-Masson (F&M) staining, and multi-photon microscopy in a pigmented 3D human skin model, MelanoDerm. However, glucose did not alter the gene expression or protein levels of major melanogenic proteins in melanocytes. While glucose potently decreased intracellular tyrosinase activity in melanocytes, it did not reduce mushroom tyrosinase activity in a cell-free experimental system. However, glucose was metabolized into lactic acid, which can powerfully suppress tyrosinase activity. Thus, we concluded that glucose indirectly inhibits tyrosinase activity through conversion into lactic acid, explaining its anti-melanogenic effects in melanocytes.

## 1. Introduction

Melanogenesis is the process of melanin production and is essential for skin protection. Melanin absorbs ultraviolet (UV) light and protects the skin from the damaging effects of UV light and free radicals [1]. However, because excessive production of melanin causes hyperpigmentation such as freckles and lentigo, which may be considered unaesthetic, much research has been dedicated towards finding effective depigmentary ingredients for cosmetics or medicines [2,3,4].

Sugar is a powerful humectant for skin moisturization and is used as a cosmetic ingredient for moisturizing skin with minimal side-effects. In addition, sugars and sugar-related agents affect melanogenesis [5,6]. Glycosylation of tyrosinase, a key enzyme involved in melanin synthesis, can be altered, inhibit its catalytic activity and accelerating its degradation [7]. The role of sugars in melanogenesis has been highlighted by studies investigating the effect of glycosylation on the pigmentation phenotype of melanocytes and the roles of sugar residues on the catalytic activity of tyrosinase [5,6,7]. Some studies have reported that sugar derivatives can inhibit tyrosinase maturation, affecting its glycosylation. For example, N-acetylglucosamine (NAG), an amino hexose produced physiologically by addition of an amino group to glucose, disrupts tyrosinase glycosylation, resulting in depigmenting effects in guinea pig skin and in human skin [8]. Additionally, recent studies have showed that sugar-related compounds inhibit tyrosinase expression or activation as well as alter its glycosylation. For example, we evaluated the whitening efficacy of galacturonic acid (GA), a sugar acid that is an oxidized form of galactose and the main component of pectin. Galacturonic acid exerts a whitening effect through regulation of tyrosinase activity and expression in B16 murine melanoma cells and a human skin equivalent [9]. In another report, a new type of cyclic oligosaccharide, known as cyclic nigerosyl nigerose (CNN), showed a weak but significant direct inhibitory effect on the enzymatic activity of tyrosinase, suggesting one possible mechanism of hypopigmentation [10]. Similar to tyrosinase maturation by proper glycosylation, the expression or activity of CNN could be a target of anti-melanogenic agents [11]. However, numerous anti-melanogenic agents have severe side-effects, such as vitiligo [12,13]. There is therefore great interest in safer depigmentary compounds.

Here, we investigated the anti-melanogenic effects of glucose on B16 murine melanoma cells and normal human melanocytes. In addition, we examined tissue color and epidermal status using tissue section staining of a human skin equivalent. Based on our findings, we propose that the whitening effect of glucose is dependent on lactic acid production, resulting in tyrosinase inactivation.

## 2. Results

### 2.1. Anti-Melanogenic Efficacy of Glucose in B16 and NHMs

To investigate the anti-melanogenic effect of glucose, we used two types of melanocytes, B16 melanoma cells (a murine melanoma cells line) and normal human melanocytes (NHMs). First, we determined if glucose was toxic to B16 cells. Glucose did not show any cytotoxicity at concentrations up to 100 mM in B16 cells, as shown in Figure 1A. Based on the cytotoxicity data, B16 cells were treated with various concentrations of glucose for 72 h in the presence of α-melanocyte-stimulating hormone (MSH), an inducer of melanogenesis. As shown in Figure 1B, glucose clearly and significantly down-regulated the intracellular melanin content in a dose-dependent manner. Kojic acid (KA) was used as a reference compound for anti-melanogenesis because it is often used as a skin lightening cosmetic ingredient [1,3,4]. The color of lysates in glucose-treated cells was lighter than the color of control cells (Figure 1C). In addition, we confirmed that the amount of melanin secreted into the culture media decreased and the color of the media clearly brightened (Figure 1D). Next, we investigated the anti-melanogenic effect of glucose on darkly pigmented NHMs. Glucose at concentrations up to 100 mM was not cytotoxic for up to for 4 days (Figure 1E). When melanin content was determined after glucose treatment of NHMs for 4 days, we found that melanin content decreased in a dose-dependent manner (Figure 1F). Taken together, these results indicate that glucose suppresses melanin synthesis in melanocytes.

### 2.2. Whitening Effect of Glucose on a 3D Human Skin Equivalent

To further define the anti-melanogenic ability of glucose, we used a pigmented 3D human skin model, MelanoDerm. As described in the Materials and Methods section, glucose was topically applied to the MelanoDerm for 18 days, and cell viability was determined by CCK-8 assay. As shown in Figure 2A, no tissue cytotoxicity was observed after glucose treatment for 18 days. Tissue color changes were assessed by photography. As shown in Figure 2B, glucose-treated tissue was lighter in color than phosphate-buffered saline (PBS)-treated tissue. In addition, hematoxylin and eosin (H&E) staining revealed that glucose did not induce tissue collapse, while fontana-masson (F&M) staining showed that glucose decreased the number of hyperpigmented melanocyte (as indicated by arrows) in the basal layer (Figure 2C).

To further investigate changes in melanin content, we compared the auto-fluorescence signals of melanin in the melanocyte layers of PBS- and glucose-treated tissues using two-photon excitation fluorescence (TPEF) microscopy, as shown in Figure 2D. An analysis of these images showed that the melanin volume was decreased by approximately 36% and TPEF signal intensity for melanin in the melanocyte-rich area was decreased by approximately 38% in glucose-treated tissues compared with PBS-treated tissues.

### 2.3. Effect of Glucose on the Expression of Melanogenic Proteins in Melanocytes

Next, to elucidate the depigmentation mechanism of glucose in melanocytes, we assessed its effect on the expression of melanogenic enzymes such as tyrosinase and Tyrp-1 in B16 cells and NHMs. B16 cells were treated with glucose for the indicated times in the presence of α-MSH. Then, protein levels of tyrosinase and Tyrp-1 were determined by Western blot assay (Figure 3A). Expression level of tyrosinase and Tyrp-1 were not decreased by glucose at any time point. Furthermore, transcript levels of tyrosinase were not affected by glucose in α-MSH-stimulated B16 cells (Figure 3B). Similar to B16 cells, protein levels of tyrosinase, Tyrp-1, and MITF were not inhibited by glucose (Figure 3C) in NHM cells treated with glucose, and mRNA levels of tyrosinase and Tyrp-1 were also not decreased, but rather slightly increased by glucose (Figure 3D).

### 2.4. Effect of Glucose on Tyrosinase Activity

To further define the action mechanism of glucose, we tested if it had an inhibitory effect on mushroom tyrosinase activity. As shown in Figure 4A, we found that glucose had no inhibitory effect on mushroom tyrosinase activity, indicating that glucose does not directly affect tyrosinase activity. We next performed a total intracellular tyrosinase activity assay using glucose-treated cell lysates from both B16 cell and NHMs. Interestingly, intracellular tyrosinase activity was clearly inhibited in a dose-dependent manner in glucose-treated B16 cells in the presence of α-MSH (Figure 4B). In NHMs, glucose also inhibited intracellular tyrosinase activity (Figure 4C). These data support the possibility that glucose indirectly inactivates tyrosinase in melanocytes.

### 2.5. Tyrosinase Inactivation by the Production of Lactic Acid in Glucose-Treated Melanocytes

Glucose is converted into the cellular metabolite lactate, which is lactic acid in solution and which has been reported to be effective in treating pigmentary lesions [14,15]. Therefore, we hypothesized that glucose is converted into lactic acid in melanocytes and that increased levels of lactic acid inhibit melanogenesis through tyrosinase inactivation. To evaluate this hypothesis, we first assessed the production of lactic acid in media from glucose-treated melanocytes. As expected, glucose significantly up-regulated the lactic acid content in B16 cells-cultured media (Figure 5A). In addition, because lactic acid is known to directly inhibit tyrosinase activity [15], we evaluated if lactic acid suppressed mushroom tyrosinase activity. As shown in Figure 5B, lactic acid dramatically inhibited mushroom tyrosinase activity, unlike glucose. These results suggest that conversion of glucose to lactic acid has an anti-melanogenic effect via tyrosinase inactivation by the lactic acid.

## 3. Discussion

Sugars or sugar-derived materials are often used as cosmetic ingredients for skin protection and physiological control. For example, raffinose increases mTOR-independent autophagy and reduces cell death in UVB-irradiated keratinocytes, indicating that the natural agent raffinose has potential value in limiting photodamage [16]. Trehalose and sucrose are novel activators of autophagy in human keratinocytes through an mTOR-independent pathway [17]. These findings provide new insight into the sugar-mediated regulation of autophagy in keratinocytes. In the case of glucose, topical glucose was shown to induce claudin-1 and filaggrin expression in a mouse model of atopic dermatitis and in keratinocyte culture, indicating that it has an anti-inflammatory effect by repairing skin barrier function [18]. In addition, glucose inhibits proliferation and enhances the differentiation of skin keratinocytes [19]. Therefore, glucose regulates various aspects of epidermal physiology, such as skin barrier functions and keratinocyte hydration levels.

Sugars can act as depigmentary agents via several different mechanisms that involve tyrosinase. For example, some agents inhibit tyrosinase maturation, while other agents induce inhibition of tyrosinase gene expression or activity [5,8,9,10]. However, few studies have investigated the mechanisms underlying the depigmentation effects of glucose.

To determine the effect of glucose on melanogenesis, we previously focused on liver X receptor (LXRs), which are ligand-activated nuclear receptors that play pivotal roles in lipid metabolism and cholesterol homeostasis [20]. We found that liver X receptor activation inhibits melanogenesis through the acceleration of extracellular signal regulated kinase (ERK)-mediated microphthalmia-associated transcription factor (MITF) degradation [21]. Furthermore, glucose is an endogenous LXR ligand [22]. Thus, we hypothesized that glucose has anti-melanogenic effects due to activation of an LXR-dependent pathway. However, as shown in Figure 3C, glucose did not alter the expression levels of tyrosinase or MITF, although LXR activation reduced the expression levels of these proteins in melanocytes. Therefore, we concluded that glucose had depigmenting effects on melanocytes independent of LXR activation.

Glucose is the principal substrate for energy production, and it is hydrolyzed via serial reactions of several enzymes, known as glycolysis. Numerous studies have shown that glycolysis has only one end product, lactic acid, whether under aerobic or anaerobic conditions. Under aerobic conditions (O_2_), lactate is utilized as the substrate of mitochondrial lactate dehydrogenase (mLDH). LDHs convert it to pyruvate that enters the tricarboxylic acid (TCA) cycle. Under anaerobic conditions (N_2_), lactate is accumulated in the cytosol. Therefore, the glycolysis pathway begins with glucose as its substrate and terminates with the production of lactate as its main end product [23]. In addition, David et al. measured the fraction of glucose that was converted to lactic acid or pyruvate in the normal human melanocyte [24]. Most of the metabolites of glucose were lactic acid rather than pyruvate.

Lactic acid is an alpha hydroxy acid (AHA); these acids are used extensively in cosmetic formulations as superficial peeling agents [25]. In addition, lactic acid suppresses melanin formation by directly inhibiting tyrosinase activity, an effect independent of its acidic nature, which means that lactic acid’s effects on pigmentary lesions are due not only to acceleration of epidermal cell turnover, but also direct inhibition of melanin formation in melanocytes [15]. Thus, we reasoned that glucose may exert its anti-melanogenic effect via lactic acid production, because glucose can be converted to lactic acid. As expected, we found that glucose treatment resulted in lactic acid production in melanocytes (Figure 5A). In addition, we confirmed that lactic acid powerfully and directly inhibited tyrosinase activity (Figure 5B). Usuki et al. also discovered that lactic acid decreased intracellular tyrosinase activity in B16 cells and human HM3KO cells [15]. In the report, mRNA and protein levels of tyrosinase and Tyrp-1 were not affected by lactic acid. Together, these data indicate that glucose increases lactic acid production and that this lactic acid directly inhibits tyrosinase activity without affecting gene expression levels, indicating that glucose has an anti-melanogenic effect in melanocytes via indirect tyrosinase inactivation dependent on lactic acid production. However, further experimentations are necessary to validate the role of lactic acid and glucose in depigmentation.

Collective data from this study provide preliminary evidence supporting the utility of topical glucose as an effective whitening as well as moisturizing reagent that can be safely used in cosmetics and medicinal formulations.

## 4. Materials and Methods

### 4.1. Materials

D-glucose, α-MSH, kojic acid (KA), L-tyrosine, L-DOPA, and lactic acid were purchased from Sigma-Aldrich (St. Louis, MO, USA). Antibodies against tyrosinase and actin were purchased from Abcam (Cambridge, UK). Antibody against Tyrp-1 was purchased from Santa Cruz Biotechnology (CA, USA). Antibody against MITF was purchased from Proteintech (city, IL, USA).

### 4.2. Cell Culture and Viability Assay

We purchased B16 murine melanoma cells, Dulbecco’s modified Eagle’s medium (DMEM), and fetal bovine serum (FBS) from the American Type Culture Collection (ATCC, Manassas, VA, USA). B16 cells were cultured in DMEM containing 4500 mg/L high glucose (ATCC 30-2002) as recommended by the manufacturer (ATCC) supplemented with 5% FBS, and incubated at 37 °C in a humidified atmosphere containing 95% air and 10% CO_2_. Darkly pigmented primary NHMs were purchased from Thermo Fisher Scientific (#C2025C; Waltham, MA, USA). Cells were cultured in Medium 254 (#M254500) supplemented with Human Melanocyte Growth Supplement (#S0025) and incubated at 37 °C under a 5% CO_2_ atmosphere. For experiments, primary NHMs between passages 4 and 7 were used. The viability of cultured cells was assessed using a Cell Counting Kit-8 (CCK-8) as described by the manufacturer (DOJINDO, Tokyo, Japan).

### 4.3. Measurement of Melanin Content

Melanin content was determined as described in previous reports [2,3,4]. Briefly, B16 cells were treated with the indicated concentrations of glucose in the presence of α-MSH (200 nM) for 72 h. NHMs were treated with the indicated concentrations of glucose for 4 d. Thereafter, all cells were washed with phosphate-buffered saline (PBS) and dissolved in 1 N NaOH at 60 °C for 1 h. Cell lysates were transferred to a 96-well plate, and absorbance was measured at 405 nm. The values were normalized based on the protein concentrations in each sample well.

### 4.4. Mushroom Tyrosinase Activity Assay

We investigated the direct effects of the indicated concentrations of glucose on mushroom tyrosinase activity. Briefly, 100 μL of phosphate buffer containing glucose was mixed with mushroom tyrosinase (10 units/well) and combined with 50 μL of 0.03% L-tyrosine or L-DOPA in distilled water. Then, the mixture was incubated together at 37 °C for 10 min, and absorbance was measured at 405 nm. Kojic acid (KA), a well-known anti-tyrosinase agent, was used as a reference compound.

### 4.5. Intracellular Tyrosinase Activity Assay

Briefly, B16 cells or NHMs were treated with the indicated concentrations of glucose for the indicated times. Then, cells were washed with PBS and lysed by incubation in 50 mM phosphate buffer (pH 6.8) containing 1% Triton X-100 and 0.1 mM phenylmethyl-sulfonyl fluoride. Cellular lysates were then centrifuged at 12,000 rpm at 4 °C for 20 min. The supernatant containing cellular tyrosinase was collected and the protein content was determined for normalization. The cellular extract was incubated with L-DOPA in phosphate buffer and dopachrome formation was monitored by measuring absorbance at 405 nm within 30 min.

### 4.6. RNA isolation and Real-Time Quantitative Reverse Transcription-Polymerase Chain Reaction (qRT-PCR)

To determine relative mRNA expression of selected genes, total RNA was isolated with TRIzol (Invitrogen, CA, USA), according to the manufacturer’s instructions, and 4 µg RNA was reverse-transcribed into cDNA using RT-premix (Bioneer, Seoul, South Korea). Quantitative PCR was performed using an ABI 7500 Fast Real-Time PCR System (Applied Biosystems, Foster City, CA, USA). The qRT-PCR primer sets for tyrosinase and Tyrp-1 were purchased from Applied Biosystems, and TaqMan Gene Expression Assay kits (Applied Biosystems) were used for amplification. Target gene expression was normalized to that of the housekeeping gene encoding ribosomal protein lateral stalk subunit P0 (RPLP0). Relative quantization was performed using the comparative ∆∆C_t_ method according to the manufacturer’s instructions.

### 4.7. Western Blotting

Cells were washed twice with cold PBS and then lysed in ice-cold modified RIPA buffer (Cell Signalng Technology, MA, USA) containing protease inhibitors (Calbiochem, La Jolla, CA, USA). Total protein concentration was determined, and the proteins were resolved by SDS-PAGE on 4–12% gradient Bis-Tris gels (Thermo Fisher Scientific, Waltham, MA, USA), transferred to nitrocellulose membranes (Thermo Fisher Scientific). After transfer, membranes were blocked in 5% blocking solution. Membranes were incubated with primary antibodies at 4 °C for 24 h, washed with Tris-buffered saline containing 0.1% Tween-20 (TBST), and exposed to peroxidase-conjugated secondary antibodies for 1 h at room temperature. Membranes were rinsed three times with TBST. Chemiuminescent signal was developed using Western blotting ECL reagent (GE Healthcare, Hatfield, UK).

### 4.8. Three-Dimensional (3D) Human Skin Equivalent

We used MelanoDerm (MEL-300-B; MatTek Corp., Ashland, MA, USA) as a human skin tissue model. This viable, reconstituted, 3D human skin equivalent was derived from black donors and contains normal melanocytes and keratinocytes. MelanoDerm was grown at the air-liquid interface in EPI-100-NMM-113 medium (MatTek Corp, Ashland, MA, USA). Prior to glucose treatment, tissues were washed with 1 mL PBS to remove residual compounds. Glucose was dissolved in PBS. Final concentration of glucose was 2%. The control sample was treated only with PBS. Glucose was applied to MelanoDerm on days 1, 4, 6, 8, 11, 13, and 15. After 18 days, MelanoDerm tissues were fixed in 4% buffered formaldehyde, embedded in paraffin, cut to a thickness of 3 μm, and subjected to H&E and F&M staining. The viability of the tissue samples was assessed using a Cell Counting Kit-8 (CCK-8) as described by the manufacturer (DOJINDO, Tokyo, Japan). Pigmentation of the MelanoDerm was assessed by comparing the change in L* value.

### 4.9. Two-Photon Excitation Fluorescence (TPEF) Imaging

To visualize the distribution of melanin in the 3D human skin equivalent, we performed TPEF imaging, as described in our previous reports [3,4]. Briefly, each MelanoDerm preparation was fixed in 4% formalin for 24 h at 4 °C, and then washed with PBS/0.1% BSA (bovine serum albumin, Merck, Branchburg, NJ, USA). The TPEF images were acquired from the basal layer to measure intracellular melanin in the melanocyte layer. Relative TPEF signal intensities for melanin in the measurement volume were quantified using Image-Pro Premier 3D software (Media Cybernetics, Inc., Bethesda, MD, USA).

### 4.10. L-lactate Assay

B16 cells were seeded at 1.0  ×  10^5^ cells per well in a 12-well plate. After treatment with glucose for 3 days, extracellular lactate levels were quantified using an L-lactate colorimetric assay kit (Abcam, ab65331, Cambridge, UK) according to the manufacturer’s protocol.

### 4.11. Statistical Analysis

Data are expressed as means ± SDs (standard deviations), and statistical significance was determined by Student’s *t*-test. A *p*-value < 0.05 was considered statistically significant.

## Figures and Tables

**Figure 1 ijms-21-01736-f001:**
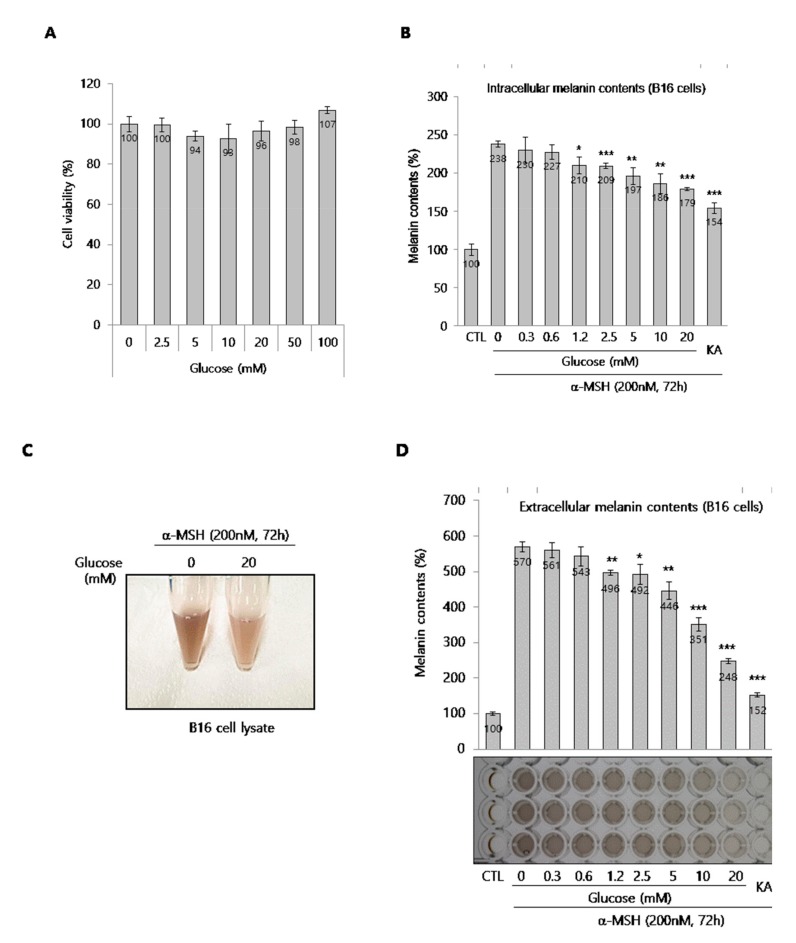
Effect of glucose on B16 cells and normal human melanocytes (NHMs). (**A**) Effect of glucose on the viability of B16 cells. (**B**) Intracellular melanin contents in α-MSH-stimulated B16 cells. Intracellular melanin contents were determined using cell lysates, as described in the methods section. (**C**) The color of cell lysate. (**D**) Extracellular melanin contents were determined using cultured media containing secreted melanin after glucose and α-MSH co-treatment for 72 h. KA indicates kojic acid (100 μg/mL), which was used as a reference compound. The photograph shows the colors of culture media. (**E**) Effect of glucose on the viability of NHMs. (**F**) Effects of glucose on melanin synthesis in NHMs. NHMs were treated with the indicated concentrations of glucose for 4 d, washed, and lysed with NaOH to determine the intracellular melanin contents. The melanin contents were estimated by absorbance at 405 nm and normalized by the total protein contents. Data are expressed as the mean ± SD of at least three independent measurements (**p* < 0.05, ***p* < 0.01, ****p* < 0.001).

**Figure 2 ijms-21-01736-f002:**
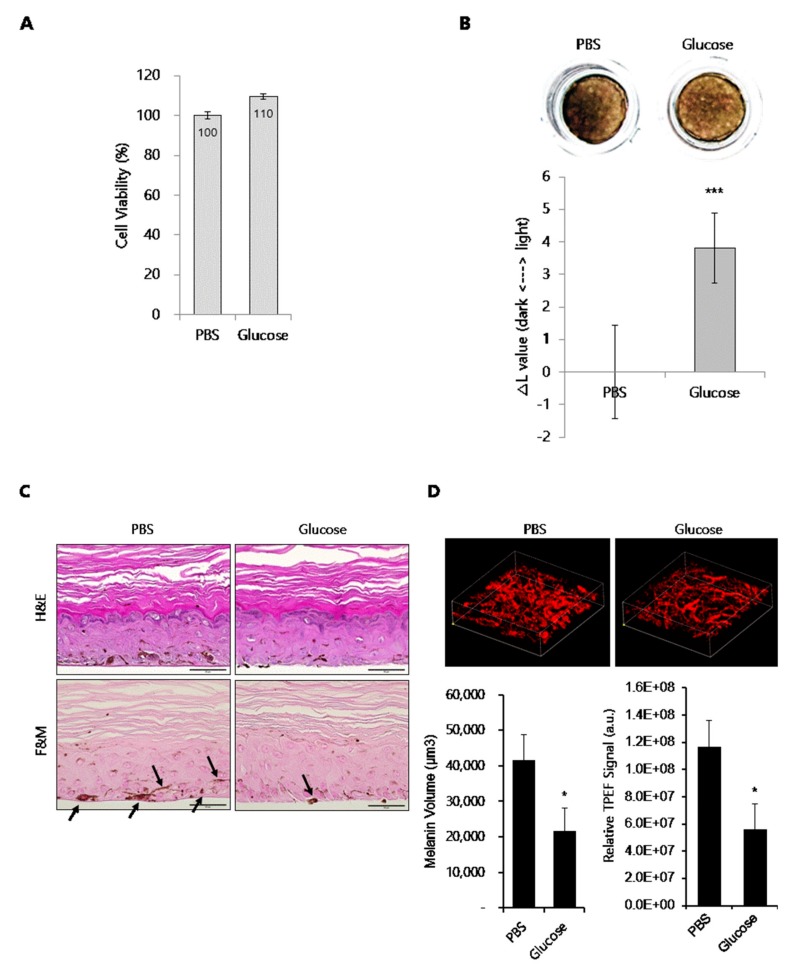
Effect of glucose on human skin equivalent, MelanoDerm. (**A**) Viability of human skin equivalents treated with glucose. (**B**) Human skin equivalents (MelanoDerm; *n* = 3) were topically treated with glucose for 18 d, and then photographed. The ΔL value indicates the degree of lightness compared with PBS-treated tissue. (**C**) H&E and F&M staining of tissue sections. The slides were fixed in formaldehyde solution and embedded in paraffin wax for staining (scale bar, 50 μm). Black arrows indicate pigmented melanocytes. (**D**) Melanin imaging (200 × 200 × 60 µm^3^) of human skin equivalents was performed using TPEF microscopy. Pseudocolored (red) signals indicate melanin (scale bar, 50 µm). The graphs indicate the quantification of melanin volume and TPEF signals. Data are expressed as the mean ± SD of at least three independent measurements (**p* < 0.05, ****p* < 0.001).

**Figure 3 ijms-21-01736-f003:**
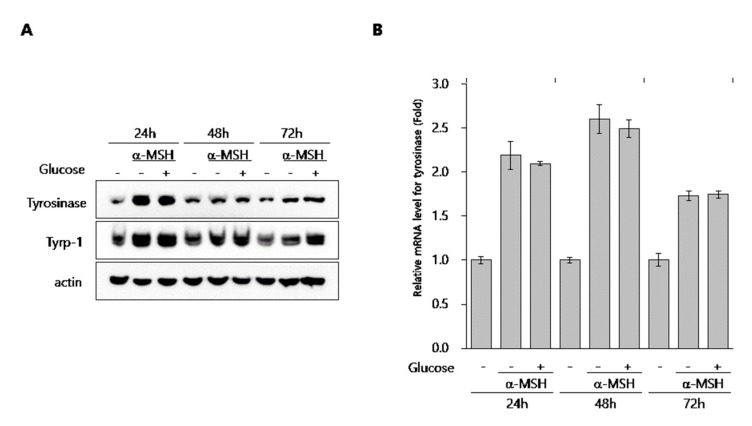
Effect of glucose on the expression of melanogenic proteins in B16 cells and NHMs. (**A**, **B**) B16 cells were treated with α-MSH for the indicated time in the presence or absence of 20 mM glucose. Then, Western blot (**A**) and qRT-PCR (**B**) assays were performed. (**C**) NHMs were treated with the indicated concentrations of glucose for 48 h. Then, the Western blot assay was performed. (**D**) NHMs were treated with the indicated time in the presence of 50 mM glucose. Then, qRT-PCR assays were performed. Data are expressed as the mean ± SD of at least three independent measurements.

**Figure 4 ijms-21-01736-f004:**
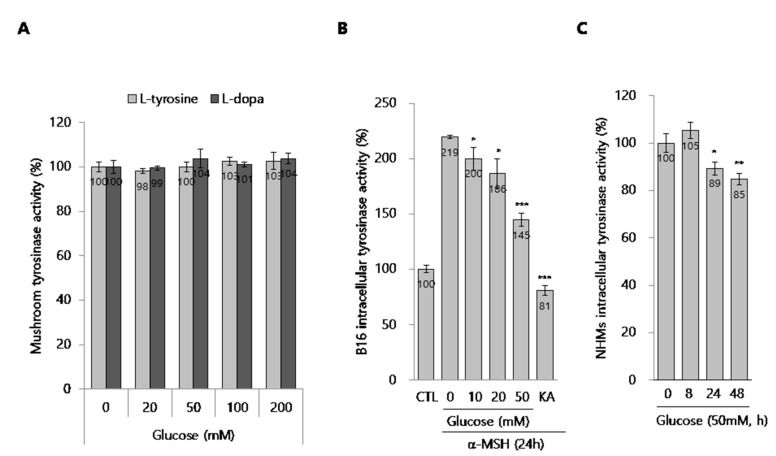
Effect of glucose on the tyrosinase activity. (**A**) Effect of glucose on the mushroom tyrosinase activity in cell-free system. (**B**,**C**) Cellular tyrosinase activity assay. (**B**) B16 cells were treated with the indicated concentrations of glucose for 24 h. Then, the cellular tyrosinase activity assay was performed, as described in the methods section. (**C**) NHMs were treated with 50 mM glucose for the indicated time. Then, the cellular tyrosinase activity assay was performed, as described in the methods section. The cellular tyrosinase activity was normalized by the total protein contents. Data are expressed as the mean ± SD of at least three independent measurements (**p* < 0.05, ***p* < 0.01, ****p* < 0.001).

**Figure 5 ijms-21-01736-f005:**
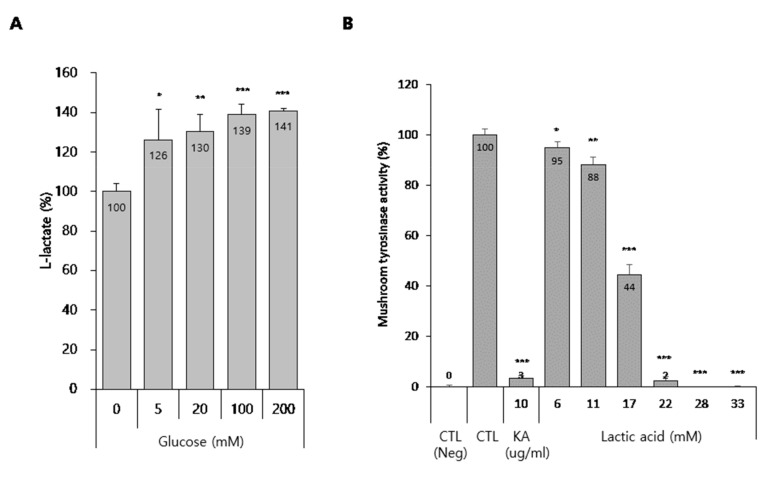
Production of lactate by glucose and effect of lactic acid on the tyrosinase activity. (**A**) Lactate production in glucose-treated B16 cells. (**A**) B16 cells were treated with the indicated concentrations of glucose for 3 d. Then, a lactate assay was performed using the cultured media, as described in the methods section. (**B**) Effect of lactic acid on the mushroom tyrosinase activity in cell-free system. Data are expressed as the mean ± SD of at least three independent measurements (**p* < 0.05, ***p* < 0.01, ****p* < 0.001).

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
