# Peer review of "Glucose Exerts an Anti-Melanogenic Effect by Indirect Inactivation of Tyrosinase in Melanocytes and a Human Skin Equivalent"

_ijms, 2020, doi:10.3390/ijms21051736_

Round 1

Reviewer 1 Report

Authors did a good job designing this study. The role of glucose, as a natural product, in skin whitening is of great interest, especially in the field of skin biology. My main comments are regarding the data presented in figures 1 and 2.   Pictures of cells (or cell pellets) should be added to the figure 1 so that the differences in pigment content would be more visible, instead of media. In figure 2C authors should provide more details about glucose inhibiting melanin. F&M staining pictures are not convincing. The same is for D. Detailed description and quantification of the results would be of more use.

Reviewer 2 Report

"The authors investigated the effect of glucose on melanin production
by melanocytes and found that glucose decreases the amount of
intracelleluar melanin, both in normal human melanocytes and skin
equivalents. Moreover, the authors observed that this effect is not
due to the modulation of tyrosinase expression or activity by glucose
but that it could rather be due to the inhibition of tyrosinase
activity by lactic acid. The manuscript is interesting and the data
convincing, but it would benefit from having the following major
issues addressed:

1 - Fig. 2 needs some sort of quantification;

2 - The authors must test the effect of lactic acid in intracellular
activity, at least in B16 cells, similar to what they did in Fig.
4B. This would provide stronger evidence to the claim that glucose
inhibits melanogenesis indirectly via tyrosinase activity inhibition
by lactic acid.

Minor issues:

1 - Authors should show the data regarding glucose cytotoxicity of up
to 4 days.

2 - It is not explained in the text the rationale for the use of
kojic acid.

3 - Why didn't the authors use KA and a-MSH in Fig. 1D, as in Fig.
1B? A similar issue applies to Fig. 4B and C.

4 - In Fig. 1C, can the authors include a color photo of the plate
instead of B&W?"

Reviewer 3 Report

The manuscript titled “Glucose exerts an anti-melanogenic effect by indirect inactivation of tyrosinase in melanocytes and a human skin equivalent” demonstrates that glucose indirectly inhibits tyrosinase activity through conversion into lactic acid, partially explaining its anti-melanogenic effects. Though this manuscript demonstrates the involvement of lactic acid, it fails to demonstrate the role of lactic acid dehydrogenase activity.

Minor points:

Increase size of figure labels Minor language corrections Please elaborate on the metabolic pathways involved in lactic acid production

Major points:

Please demonstrate control experiments using inhibitors of lactic acid dehydrogenase Please demonstrate the preference of anaerobic metabolism in these experiments and the lack of aerobic pathway

Round 2

Reviewer 2 Report

The authors addressed all the comments made by the reviewers by performing new experiments and editing the text. As a result, the manuscript has been significantly improved. However, a few minor issues remain:

1 - Are the differences in Fig. 1E non-significant? Actually, the authors should always refer explicitly when the differences are non-significant, but specially in this case, where there is a noticeable decrease in cell viability with 250 mM glucose.

2 - In Fig. 2D there are no error bars and no reference to how many times the experiment was repeated.

3 - Regarding the use of kojic acid, the authors should at least refer in the text what they wrote in the rebuttal letter, to help the non-expert readers.

4 - The issue regarding the inhibition of tyrosinase activity by lactic acid is not fully resolved. The authors refer that they have "tested the effect of lactic acid in intracellular tyrosinase activity using B16 cells" but they do not say or show what was the result. It is true that this experiment was published (in ref. 26 and not 25), but if so, the authors must acknowledge this, for example in the discussion, referencing correctly the published study.

Author Response

Thank you for reviewer's comment.

1 - Are the differences in Fig. 1E non-significant? Actually, the authors should always refer explicitly when the differences are non-significant, but specially in this case, where there is a noticeable decrease in cell viability with 250 mM glucose.

Response : As the reviewer mentioned, the differences are significant between control group and glucose (250 mM)-treated group. We removed data for ‘250mM glucose’ and corrected the figure 1E and 1F in the revised manuscript.

2 - In Fig. 2D there are no error bars and no reference to how many times the experiment was repeated.

Response : In Fig. 2D, we showed the representative graphic data and the graphs was produced by the graphic data. However, we re-calculated and produced two graphs through three measurements. In the revised manuscript, we added the data and statements.

3 - Regarding the use of kojic acid, the authors should at least refer in the text what they wrote in the rebuttal letter, to help the non-expert readers.

Response : As the reviewer mentioned, we added the explanation for kojic acid with the red text in the revised manuscript.

4 - The issue regarding the inhibition of tyrosinase activity by lactic acid is not fully resolved. The authors refer that they have "tested the effect of lactic acid in intracellular tyrosinase activity using B16 cells" but they do not say or show what was the result. It is true that this experiment was published (in ref. 26 and not 25), but if so, the authors must acknowledge this, for example in the discussion, referencing correctly the published study.

Response : As the reviewer mentioned, we added the further explanation with the red text and corrected ‘reference number’ in the discussion section of the revised manuscript.

Reviewer 3 Report

The authors have addressed the reviewer’s comments and suggestions satisfactorily. However it is recommended that a statement be included in the conclusions that further experimentation is necessary to validate the role of lactic acid in de-pigmentation.

Author Response

Thank you for the reviewer’s comment. As the reviewer mentioned, we added the sentences in the discussion section of the revised manuscript.